# Environments, risk and health harms: a qualitative investigation into the illicit use of anabolic steroids among people using harm reduction services in the UK

Andreas Kimergård,[1,2] Jim McVeigh[2]

For numbered affiliations see end of article.

**Correspondence to**
Andreas Kimergård;
Andreas.Kimergard@kcl.ac.uk

## ABSTRACT

**Objectives:** The illicit use of anabolic steroids among the gym population continues to rise, along with the number of steroid using clients attending harm reduction services in the UK. This presents serious challenges to public health. Study objectives were to account for the experiences of anabolic steroid users and investigate how 'risk environments' produce harm.

**Methods:** Qualitative face-to-face interviews with 24 users of anabolic steroids engaged with harm reduction services in the UK.

**Results:** Body satisfaction was an important factor when deciding to start the use of anabolic steroids. Many users were unaware of the potential dangers of using drugs from the illicit market, whereas some had adopted a range of strategies to negotiate the hazards relating to the use of adulterated products, including self-experimentation to gauge the perceived efficacy and unwanted effects of these drugs. Viewpoints, first-hand anecdotes, norms and practices among groups of steroid users created boundaries of 'sensible' drug use, but also promoted practices that may increase the chance of harms occurring. Established users encouraged young users to go to harm reduction services but, at the same time, promoted risky injecting practices in the belief that this would enhance the efficacy of anabolic steroids.

**Conclusions:** Current steroid-related viewpoints and practices contribute to the risk environment surrounding the use of these drugs and may undermine the goal of current public health strategies including harm reduction interventions. The level of harms among anabolic steroid users are determined by multiple and intertwining factors, in addition to the harms caused by the pharmacological action or injury and illness associated with incorrect injecting techniques.

## INTRODUCTION

There is growing interest within public health in understanding how dynamic relationships between people who use drugs and the surrounding environment can affect the production or reduction of drug-related harm.[1] In this approach, focus changes from individual-level factors alone to social situations and environments embedding the use of drugs. The present study explores the illicit use of anabolic steroids for the purpose of human enhancement among the gym population in the UK. The emphasis is that harm among users is contingent not just on pharmacologically active substances in 'anabolic steroid' products but also on the 'risk environment'—a space where multiple factors (physical, social, economic, policy) intertwine to affect the chance of harm occurring.[2]

There has been reporting of harms relating to anabolic steroids, including adverse reactions such as acne, hair loss, gynaecomastia, disruption of growth, damage to tendons and ligaments, testicular atrophy, erectile dysfunction, liver damage (especially with oral products) and cardiovascular events.[3] It has also been reported in the literature that anabolic steroids may cause aggressive behaviour.[4] Severity of these effects would appear to be dose dependent.

Additional harms relate to the route of administration given that many users inject at least some of their drugs,[5–7] leading to

concerns regarding incorrect injection technique and transmission of blood borne viruses through the sharing of injecting equipment.[8] A recent study of anabolic steroid injectors in England and Wales, reported that 8.9% had shared injecting equipment.[6] Of considerable concern for public health is the identification of 1.5% being HIV positive, 9% with antibodies to the hepatitis B core antigen (anti-HBc) and 5% to hepatitis C virus (anti-HCV). In the case of hepatitis B, biological samples were only tested for core antigen identifying those that had been exposed to hepatitis B, rather than the level of users who are carriers of the virus. Still, it is noteworthy that the results of the study indicate comparable levels of HIV prevalence between injectors of anabolic steroids and intravenous heroin injectors in England and Wales.

**Strengths and limitations of this study**

- Advances our understanding of the risks posed by the use of anabolic steroids by exploring the relationship between harm in individual users and the surrounding environment.
- This study recruited a broad cross section of anabolic steroid users, proving access to accounts of the experiences of this group of drug users.
- Findings reflect those users attending harm reduction services. Users who choose not to engage with these services may have a different risk profile.

Studies in the USA and in Sweden suggest that steroid use may be associated with the use of recreational drugs.[9] [10] In the UK, prevalence of cocaine use among steroid users is substantially higher than the comparable general population.[6] However, large scale studies with an in-depth focus on the causal relationship between the use of steroids and recreational drugs are required.

The complex polypharmacy adopted by many users of these drugs poses additional risks. Common regimens include the concurrent usage of various drugs for enhancement purposes including a range of anabolic agents, stimulants and an array of drugs used as self-treatment of steroid-induced side effects[5–7] [11] through to the use of dietary and herbal supplements that may be contaminated with undeclared harmful substances.[12] [13]

While accounts of such potential health harms associated with anabolic steroids advance the understanding of the direct harms posed by various active substances, along with their route of administration, they offer little insight into the social processes and environmental factors exhibiting relations of risk and enablement.[1] Empirical qualitative studies show that users exchange anabolic steroid-related information on methods to reduce or avoid adverse effects.[14] As a consequence, many users ignore or perceive the risk of steroid usage as relatively minor.[7] [15–17] Steroid users who seek the support of peer users but avoid social censure by keeping their use of drugs a secret from others may also serve as a mechanism that promotes harm.[18] By contrast, shared norms and practices in local groups of steroid users may also establish lines between what is seen as 'sensible' drug usage and what is not. One study found that the use of ephedrine (a stimulant) and nalbuphrine (an opioid) for enhancement purposes were seen by some steroid users as irresponsible risk taking because these drugs were compared to amphetamines and heroin, respectively.[19] This article reports on findings of an interview study conducted to explore the experiences of anabolic steroid users and investigate microenvironments and macroenvironments relating to physical and social risks in this particular population of people who use drugs. In addition to what is currently known about the hazards posed by the pharmacotoxicological actions of anabolic steroids, this perspective can be used to identify additional and complex risk patterns in steroid users.

## METHOD

This article is based on qualitative interviews with 24 users of anabolic steroids conducted between 2009 and 2011. The research was part of a study of harm reduction services in areas of England and Wales available to users of anabolic steroids. Such individuals have been recognised as a growing client population in harm reduction services across the UK.[8] [20]

### Recruitment

A mix of purposive and convenience sampling were used in order to select a cross section of steroid users.[21] Seven steroid users were recruited from fixed-site needle and syringe programmes; three from an outreach service in gyms; nine from steroid clinics providing harm reduction interventions designed specifically for users of anabolic steroids; one from a gym and, four offenders from a prison. Staff in harm reduction services approached steroid users and enquired if they were willing to participate with interviews taking place in a private room within these facilities. Further, a gym manager approached gym members and informed them of the research. In this case, the interview was conducted in the locker room out of earshot from other members. Offenders agreed to partake after they had been informed about the study by the head of the physical education department. These interviews were conducted without the presence of prison officers to allow for offenders to speak confidentially. The purpose of recruiting from different venues was to see whether there were any differences between steroid users attending harm reduction services and users recruited at other venues; however, it emerged that all respondents had been in contact with such services at some time point and interviews did not reveal any significant differences.

### Sample characteristics

The oldest respondent was 61 and the youngest 21 (mean age 34 years). All respondents reported previous or current use of anabolic steroids and nine reported the use of synthetic growth hormones. The majority of respondents had begun their use of anabolic steroids around the age of 25, although three had begun at the age of 16. The sample was relatively experienced as many had taken a number of steroid cycles (typically consisting of courses of drugs ranging from 6 to 12 weeks followed by a period of abstinence). The sample comprised men only, for which there are several explanations:

▶ Research shows that the majority of steroid users are men.[5] [6] [22]
▶ There is a stigma associated with female anabolic steroid use, at least in part, due to the significant masculinising effect of steroids and the societal expectations of gender identity.[23] [24]

▶ Four respondents were recruited from a prison exclusively for men.

## Data generation

Interviews were conducted by AK, lasting from 30 to 75 min. Following the collection of basic demographic information, interviews were semistructured covering the following topics based on available literature (1) body (dis)satisfaction; (2) motivations for steroid usage; (3) patterns of steroid use, sources of steroid-related information and accounts of health harms and (4) users' experiences with harm reduction services.

During the collection of data, the interviewer identified himself as a public health researcher conducting a study about the use of and policy towards anabolic steroids. It was made clear to respondents that the interviewer had no affiliation to either health or criminal justice services and that their anonymity was guaranteed. Furthermore, many harm reduction services were located in a different area than the institution the interviewer was affiliated with, this was made clear to respondents, which most likely made steroid users consider the interviewer as an 'objective outsider'. It is believed that this facilitated a trusting and honest interaction, with open sharing of personal accounts.

Most interviews were recorded, and subsequently transcribed by AK, except when prison regulation prohibited the use of electronic devices, or when respondents felt uncomfortable about discussing sensitive issues relating to anabolic steroids. In such cases, a written record was compiled immediately after the interview. Written interview records were subjected to thematic content analysis to identify and verify main themes (accounted for below).[25] The analytical process conducted as the study progressed was led by AK with input from JM in order to conduct internal checking by comparing results.[26] This process continued until themes provided explanatory accounts of the data. University ethical approval was obtained.

## RESULTS

A significant finding across all accounts was that a predominant reason to work out and take anabolic steroids was to enhance physical appearance, resonating with previous research suggesting that body dissatisfaction among men can lead to a desire to increase muscle mass leading to the use of anabolic steroids.[17 27] Of importance here is that steroid users typically expressed that there was a limit to what could be achieved 'naturally'; that is training without using steroids, and that this belief appeared to have a direct influence on decisions to use drugs:

> I knew I had pushed my body to the limit of what I naturally could and I wanted to take that a little bit further. (Steroid user aged 35)

Of equal importance is the increased body satisfaction when other people took notice of the gains that they had made, highlighting that societal norms regarding the 'perfect body' create an environment which may affect the continuation of steroid use in some people:

> People who hadn't seen me since school would come up to me and go, 'Oh my God, you look massive, you look fantastic', and it's a big hype. It's stupid because it's what is inside that counts, how you are as a person, but there is no way around it, if someone comes up to you and says, 'You look fantastic', 'You look good', 'You used to be skinny, but look at the size you have now', it's an amazing thing, and to think that I have done that myself by training hard. Because I train hard. (Steroid and growth hormone user aged 36)

In addition to this dominant theme, two broad themes emerged from the analysis:

1. Working through words and phrases, made during initial reading of the interview records, resulted in a list of categories, including: safety of illicit market products, contents of illicit market products, lack of information about illicit market products, changes regarding the illicit market (over time), trust of illicit suppliers and the use of illicit market products to determine their effects (selected examples). Based on a refinement of these categories, a reduced list of categories was compiled: uncertainties relating to the illicit market, viewpoints towards illicit suppliers and practices relating to the use of illicit market products. This led to the final coding framework, resulting in the following theme: the dangers of using drugs from the illicit market, which were unknown to many whereas others had developed practices to negotiate hazards posed by these drugs (which were in themselves creating risk).

2. Refinement of initial analysis created another list of categories: steroid users warn users about health harms, users encourage needle and syringe programme attendance, different steroids are perceived as 'safe' versus 'unsafe', users promote certain injecting practices and users rely on information from peer users (selected examples). This list was further reduced, providing categories for the initial coding framework: users promote health among users, risk relating to dissemination of information among users and promotion of steroid-related practices among groups of users. From here the final coding framework was derived, resulting in the following theme: the social structures pertaining to steroid use, which enable the transfer and dissemination of anabolic steroids-related information of dubious accuracy and validity.

This article describes these two themes, neglected to date in this area of research.

## Perceptions of illicit market products

This and other studies[22 28] have found that products claiming to contain anabolic steroids are readily accessible from the illicit market. Yet the issue of

contaminated, adulterated and misbranded drugs, along with its impact on the behaviour of users, remains a neglected area of study.

Whereas a few of the respondents expressed uncertainties pertaining to the composition and sterility of drugs obtained from the illicit market, the majority remained ignorant, or at the very least, unconcerned, regarding the associated risks. However, to other users in this study, the implication was that it was difficult for them to know exactly what pharmacologically active substance(s) was in the vials they had obtained, as well as the amount of active ingredients. This undermined their ability to plan out their regimes:

> The biggest change I've seen in steroids is the fact that when I started using them [in the late 1980s], everything you got was pharmaceutical grade. It was from a chemist or a pharmacy. It was brought in from abroad or it was from this country. They were genuine Organon, Bayer, all these famous drug companies and they all came packaged and you knew exactly what you were using. It's completely changed now. You can't get anything pharmaceutical grade, everything is 'underground'. (Steroid user and gym owner aged 45)

Another uncertainty was the confusion caused by anabolic steroids sold under fictitious product names such as 'Cutstack'. This seemed to have direct impacts on the process of obtaining information about specific types of anabolic steroids:

> Certain steroids on the [illicit] market are called different names to what they are called in the book [the so-called 'steroid handbook' by William Llewellyn,[29]] and some steroids aren't in the book. So, I go on the Internet for that information, but each site says different things about it. (Steroid user aged 22)

Of note is that these insecurities did not seem to deter users from purchasing and using such products. However, it did appear to lead to various practices being adopted in an attempt to mitigate these uncertainties. Significantly in this respect, a number of steroid users said that getting their hands on drugs of a 'sound quality' came down to finding 'someone that could be trusted' and hence many preferred to buy from 'local dealers' rather than online retailers (steroid user and competing bodybuilder aged 45). In many cases, users would dismiss speculation regarding the quality and safety of the drugs because they trusted the person who had sold them:

> I've always used the same guy, and I never had no issues with him. I always use the same person. I never had bad 'gear' from him. (Steroid user aged 22)

However, given that users in many cases cannot determine accurately the quality of such products, illicit manufacturers and suppliers are able to use this to their advantage[12]; for example, through the deliberate misbranding of vials to increase their profit. This was of concern to some:

> I've heard horror stories about dealers that I talk to… there'll be a tray of vials and they'll ask, 'what are you after?' And they say, 'well there's the bottles, there's the labels, take whatever labels you need and just stick them on the bottles.' There might be some active ingredient in them but you just can't know for sure what you're getting. (Steroid user and gym owner aged 45)

Distinct to this group was that many respondents described practices relating to self-experimentation to gauge the effects of anabolic steroids. In fact, self-experimentation was widely accepted as an approach to gain first-hand experience of the effects of these drugs:

> I think it takes a couple of courses of steroids to be able to understand how they work properly…the first time I took steroids I probably accumulated a lot more body fat than I had anticipated, but it's all part of a learning process. (Steroid user aged 35)

This approach was used on products from the illicit market to see what the effect would be and hence to assess their quality:

> Maybe you could put counterfeit stuff in a normal bottle. You wouldn't know, but you do find out the difference after you've finished your course and you are not getting the results you should be getting. I think that's a risk you take, it comes in life, it comes everywhere. (Steroid user aged 27)

These accounts highlight that for many the main concern regarding an unregulated market with products of dubious provenance is either being 'ripped off' or at the very least failing to make the desired gains. The potential for serious adverse health consequences due to poor manufacturing of products is not generally perceived as a major issue.

### Peer users and steroid knowledge: risk reduction?

In describing their experiences of anabolic steroids, it became clear that the practices, viewpoints and experiences of other users significantly affected risk behaviour, both positively and negatively. It seemed that experienced users would act as a form of regulator within the gym population; for instance by providing steroid-related information, advising users not to adopt the 'high-dose' regimes used by competing bodybuilders and persuading young people not to take steroids until they were older:

> I've always told teenagers that have asked me if they can take steroids and stuff, that it's at your own risk. You might be 5′ 7″ now, that might be as tall as you're going to get [given that anabolic steroids may shunt the normal pattern of growth,[8]] but if you don't take them, you might be 5′ 10″. That's always been my best plan and

that works. Because they think, 'Oh shit, I don't want to end up 5′ 7″ when I could be taller'. (Steroid user and gym owner aged 45)

In addition, a number of experienced steroid users told how they encouraged younger users to go to harm reduction services such as needle and syringe programmes to obtain sterile injecting equipment in order to prevent the transmission of blood borne viruses. Overall, belonging to peer and social groups of steroid users meant having access to and being influenced by information and advice with the potential to reduce the harms of steroid use. In these ways, norms and practices within environments may serve to establish boundaries by constructing 'sensible' and 'responsible' drug usage.

By contrast, viewpoints, shared norms and perceived wisdom of users also appeared to have the potential to increase risk. The widely accepted anabolic steroid 'knowledge base', passed on from user to user, is largely derived from informal self-experimentations, along with users' interpretations of the 'literature' which, in many cases, was restricted to anecdotes and peer accounts of their respective self-experimentation. In instances where robust scientific evidence was considered this was normally via a third party's interpretation of findings. Here scientific studies, including animal or in vitro experiments, with limited findings can be wildly extrapolated or findings 'cherry-picked' to support existing beliefs or rationalise perceived effects. Even so, in the group of steroid users in this study, there was a general sense of 'trust' in advice from peers:

> I go to the gym, that's where everyone goes to train. So for everyone who's actually doing the same thing, they're all in one spot. You can get all of the information you need. Why shouldn't I trust them? Because if you look at them, and think, 'well he must be doing something right because look at the shape of him'. They look really really good so why not trust them? (Steroid user aged 27)

The importance of this knowledge was clear, as steroid users told how they used it to decide which types of drugs and dosages to use. It was also used to distinguish between what types of anabolic steroids were considered 'safe':

> I have heard 'Deca' [Deca-Durabolin®; nandrolone decanoate] and 'Sustanon' [Sustanon® 250; containing a testosterone blend] to be the safest steroids there are, whether that is true I don't know. I have heard it off books, off people, off the Internet. (Steroid user aged 23)

The 'proper' use of anabolic steroids, was considered to entail variations of high doses of multiple types of anabolic steroids, provided the regimen complied with the current commonly held beliefs of synergistic efficacy, typically referred to as 'stacking'. Moreover, the use of ancillary drugs to treat or prevent steroid induced side effects was seen as an important aspect of 'responsible'

steroid usage. In fact, the use of ancillary drugs was perceived as the 'sensible' choice, rather than the reduction of steroid dosages to avoid side effects in the first place. Again, it was clear that self-experimentation was an essential part of the process:

> I know personally when my body requires hCG [human chorionic gonadotropin; taken by steroid users to trigger the production of testosterone in the body at the end of a steroid course]…It's hard to explain, but, for example, if you have problems with erection and things like that, then I know personally when my body needs hCG and that's usually when I take it. I know from personal experience if I need it or not. (Steroid user aged 35)

Shared guidelines among steroid users also extended to the administration of these drugs. On the one side users described injecting practices which were broadly in keeping with harm reduction messages pertaining to intramuscular injections, such as injection into the large muscle groups like the gluteus maximus (buttocks) or vastus lateralis (thigh) to promote diffusion and minimise the risk of abscess formation.[5] The rotation of injection sites to avoid damage and complications was also outlined. However, injections in the smaller muscles, such as the triceps brachii (back of the arms), were also promoted among users as ways to achieve a 'direct effect in the specific muscle' (steroid user aged 27)—referred to as 'spot injections' among steroid users. There is no evidence to support this action and the likelihood of injection-related complications is increased. This practice highlights the 'conflicts' between information disseminated among users (promoting the perceived efficacy of 'spot injections') and the contradictory harm reduction message from needle and syringe programmes (injecting only into large muscle groups).

Read together, multiple factors influenced steroid-related behaviour in this group of users, with peer influence being mentioned in most cases. Some appeared to have a positive influence on the risk posed by this behaviour. However, it was clear that being part of the steroid using culture ultimately seemed to create a sense of safety among steroid users regarding their use of these drugs, which appears to be misplaced.

## DISCUSSION
### Main findings, strengths and study's limitations
The overarching finding of this study was that bodily ideals, the structure, availability and function of the illicit market, accessibility of information through both local and global information networks (in gyms or via the Internet), along with steroid users' perceptions of the veracity and value of this information, can increase the chance of harm occurring in this specific drug using population.

Despite attempts among steroid users to gauge the efficacy and hence 'quality' of anabolic steroids obtained from the illicit market it is impossible for users to know

exactly what substances they are taking,[28] [30] impeding the successful self-management of risks in users. The findings suggest a need to raise awareness regarding the potential health consequences of adulterated drug usage, not only in harm reduction services but also among generic health professionals who may come into contact with steroid users. Steroid users retrieved information about the use, effects and route of administration of anabolic steroids from a number of sources including peer users within the gyms in the local environment and the Internet, a practice exhibiting risk as well as harm reduction. Information regarding anabolic steroids was used within a strategy of risk management which, in turn, provided a sense of safety in users. This is broadly in keeping with previous work highlighting that the planned use of anabolic steroids, as opposed to the 'abuse' of these drugs, was used as a strategy to avoid or minimise harm.[14] However, important uncertainties relating to this information undermines attempts to manage the risks associated with these drugs. It may be that as a consequence of the multiple sources, and sheer volume, of data about anabolic steroids, in particular on the Internet, users become less critical about the accuracy of this information.[31] This suggests that unambiguous advice is required. However, without control of the illicit market and the development of a robust evidence base this will not be possible.[3] [8]

The strength of the study was the recruitment of a cross section of steroid users, providing access to accounts of their perceptions and experiences. The collected data has added further insights into the environments and risk associated with the use of anabolic steroids. Such insights are of importance in the planning and delivery of public health initiatives. A limitation of this study is that all respondents had been in contact with harm reduction services. There is currently insufficient data to describe accurately potential differences between different groups of steroid users for example in terms of risk patterns, however, those attending harm reduction services presumably represent a group of more cautious drug users, particularly with regard to injecting behaviour, compared to 'hidden' populations of users who avoid contact with health services. A survey conducted among members of three gyms in the South Wales Region, that were reported to be popular venues for steroid using gym goers, found that 20% of participating steroid users revealed syringe sharing with other users whereas 86% reported to never having had a medical check.[32] Although there is no mention of harm reduction service attendance in this sample this might possibly be an example of a group of steroid users engaging in more risky behaviour than the sample studied here. That being said, all respondents in this present study clearly stated that they relied predominantly on the dissemination of information among peer groups of steroid users to inform their drug-taking behaviour suggesting that findings of this study extend beyond users in contact with harm reduction services.

## Implications for harm reduction practice

Harm reduction service workers are faced with the challenge of delivering advice, education and interventions in ways that will result in a reduction of overall harm even though steroid users are heavily affected by physical and social risk environments that may, in many cases, increase the chance of harm occurring; a point which came across in this study.

As a greater number of steroid using clients access harm reduction services, specific knowledge and resources will be required to deliver effective harm reduction interventions for this group of drug users.[20] Significantly in this respect, this study found that whereas all steroid users had obtained sterile needles and syringes from services only about half of them, predominantly those gaining access to steroid clinics, reported seeking additional information from these programmes with few reporting that they sought information about how to use anabolic steroids in the least harmful way. Here, the availability of knowledge within groups of steroid users seemed to act as a barrier to information from health authorities, which resonates with previous findings that health services are not usually seen as credible sources of information.[7] [14] [16] [22] Services must identify more appropriate ways of providing credible information that will be valued and result in positive behaviour change. The two current foci of user information, the internet and the experienced steroid user within the gym culture, should be further explored as methods of engagement and dissemination of harm reduction information.

It is possible that having harm reduction service providers with specific knowledge about the issues relating to the use of anabolic steroids and other drugs used for human enhancement purposes would increase the chances of delivering effective health intervention and hence challenge some of the hazardous behaviour described here. Harm reduction services have the ability to identify and respond to novel drug trends, such as the increasing number of clients using drugs for human enhancement including anabolic steroids, synthetic growth hormones and new products such as the skin tanning drug melanotan II.[12] However, the mainstay of most harm reduction services is needle and syringe distribution, along with information on safe injecting. While this approach aims to reduce the transmissions of blood borne viruses from the sharing of injecting equipment it does not help protect steroid users from a range of health harms caused by adulterated and substandard drugs, suggesting a need to focus on harm reduction messages relating to the use of products from the illicit market.

## CONCLUSIONS

The potential harms of anabolic steroids are not only the result of their pharmacological active substances or route of administration, but also the result of numerous

factors that are situationally and structurally dependent on the environment in which the use of anabolic steroids occur. These include the acceptance of using drugs from the illicit market as well as relying on uncertain information from peers. Current harm reduction interventions face barriers caused by viewpoints, norms and practices among steroid users. Strategies in public health are currently needed that are capable of functioning within existing environments, in which steroid users participate, in order to promote and protect health.

**Author affiliations**
[1]Addictions Department, National Addiction Centre, Institute of Psychiatry, King's College London, London, UK
[2]Centre for Public Health, Liverpool John Moores University, Liverpool, UK

**Acknowledgements** The authors thank the individuals who participated in this study.

**Contributors** AK conceptualised the study and conducted the interviews with support from JM. AK led the analysis which was discussed with JM as the study developed. Both authors contributed to the preparation of the paper and approved the final version. Writing was led by AK.

**Funding** This research received no specific grant from any funding agency in the public, commercial or not-for-profit sectors.

**Competing interests** None.

**Ethics approval** Liverpool John Moores University Research Ethics Committee.

**Provenance and peer review** Not commissioned; externally peer reviewed.

**Data sharing statement** No additional data are available.

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
