## [Reviewer comments · BMJ Open]

Some articles will have been accepted based in part or entirely on reviews undertaken for other BMJ Group journals. These will be reproduced where possible.

ARTICLE DETAILS

TITLE (PROVISIONAL)	Environments, risk and health harms: a qualitative investigation into the illicit use of anabolic steroids amongst people using harm reduction services in the United Kingdom
AUTHORS	Kimergård, Andreas; McVeigh, Jim

VERSION 1 - REVIEW

REVIEWER	Harrison G Pope Jr., M.D. McLean Hospital/Harvard Medical School
REVIEW RETURNED	14-Apr-2014

GENERAL COMMENTS	This is a useful contribution because it provides a window into the culture of anabolic-androgenic steroid users – an issue that is often neglected in most scientific papers regarding this form of substance abuse. Although the conclusions are obviously limited by the qualitative nature of the study, the authors acknowledge this limitation. I have only a few comments. First, the authors appropriately note that the findings are based on interviews of users attending harm reduction services, and note that individuals not engaging in such services may have a different risk profile. This is an important point, and perhaps deserves even a couple of additional sentences of discussion. Specifically, individuals attending harm reduction services presumably represent a group biased in favor of more prudent and cautious drug users, and thus the overall population of steroid users may engage in even more risky behavior, on average, than the sample studied here. On page 4, the authors mention that 9% of anabolic steroid injectors exhibited antibodies to hepatitis B core antigen. However, this piece of information by itself is not sufficiently informative. Everybody who has been exposed to hepatitis B will develop antibodies to the core antigen, and about 90% of such individuals will also develop antibodies to the surface antigen (anti-HBs). Individuals who develop both types of antibodies will recover from hepatitis B and be immune to it thereafter, whereas the small percentage of individuals who develop ONLY antibodies to the core antigen will fail to clear the virus from their systems and will become carriers of hepatitis B. Therefore the question of greatest interest is how many steroid users have antibodies exclusively to the core antigen and hence are carriers of hepatitis B, capable of infecting other people. Do the authors possess data in response to this question? Also on page 4, the authors mention that studies in the United States and in Sweden have suggested that steroid use may be associated with a progression to recreational drugs. I would suggest
---

	changing the sentence to say simply that steroid use "may be associated with use of recreational drugs," because the direction of progression can occur in both directions (i.e. steroid users may progress to use drugs such as opiates, and recreational drug users may progress to use steroids).
--	--

REVIEWER	Azenildo M. Santos, PhD Laboratório de Sociologia do Esporte, Universidade Federal de Pernambuco, Brazil
REVIEW RETURNED	29-Apr-2014

GENERAL COMMENTS	The study is very well designed, is up to date with what occurs in other countries and considers all points to reproduce this kind of qualitative study. The authors have pointed out the limitations and the objectives and discussion are clear. The references are good, as is the text-list. The methods were described properly and the sample well designed. The supplementary report is okay! Finally, in my opinion, the manuscript is "good to go". In other words, accept!
--

VERSION 1 – AUTHOR RESPONSE

Reviewer:

I have only a few comments. First, the authors appropriately note that the findings are based on interviews of users attending harm reduction services, and note that individuals not engaging in such services may have a different risk profile. This is an important point, and perhaps deserves even a couple of additional sentences of discussion. Specifically, individuals attending harm reduction services presumably represent a group biased in favor of more prudent and cautious drug users, and thus the overall population of steroid users may engage in even more risky behavior, on average, than the sample studied here.

Responds:

We agree. We have provided an examples of these potential differences in the section covering the limitations of the study.

Reviewer:

On page 4, the authors mention that 9% of anabolic steroid injectors exhibited anybodies to hepatitis B core antigen. However, this piece of information by itself is not sufficiently informative. Everybody who has been exposed to hepatitis B will develop antibodies to the core antigen, and about 90% of such individuals will also develop antibodies to the surface antigen (anti-HBs). Individuals who develop both types of antibodies will recover from hepatitis B and be immune to it thereafter, whereas the small percentage of individuals who develop ONLY antibodies to the core antigen will fail to clear the virus from their systems and will become carriers of hepatitis B. Therefore the question of greatest interest is how many steroid users have antibodies exclusively to the core antigen and hence are carriers of hepatitis B, capable of infecting other people. Do the authors possess data in response to this question?

Responds:

This is a very good point. Unfortunately, we do not have any further data on the matter. However, we

have changed the section to make it clear to readers that there are limitations of the work of Hope et al.

Reviewer:

Also on page 4, the authors mention that studies in the United States and in Sweden have suggested that steroid use may be associated with a progression to recreational drugs. I would suggest changing the sentence to say simply that steroid use "may be associated with use of recreational drugs," because the direction of progression can occur in both directions (i.e. steroid users may progress to use drugs such as opiates, and recreational drug users may progress to use steroids).

Responds:

Done.